# Association between Meningococcal Meningitis and Santa Ana Winds in Children and Adolescents from Tijuana, Mexico: A Need for Vaccination

**DOI:** 10.3390/tropicalmed8030136

**Published:** 2023-02-23

**Authors:** Enrique Chacon-Cruz, Erika Zoe Lopatynsky-Reyes

**Affiliations:** 1Institute for Global Health, University of Siena, 53100 Siena, Italy; 2Department of Pediatric Infectious Diseases, General Hospital of Tijuana, Tijuana 22010, Mexico; 3School of Family Medicine and Public Health, University of California, San Diego, CA 92093, USA

**Keywords:** meningococcal meningitis, climate, children, Santa Ana winds, meningococcal vaccine

## Abstract

Background: Based on previous studies (regional and national), Tijuana, Baja California, Mexico (across the border from San Diego, California, USA), has been shown to have the highest rate of meningococcal meningitis (MeM) in the country. However, the reason for this high incidence has not yet been established. To explain this regional/endemic public health problem, we aimed to evaluate whether there is a climatic association with MeM in the region. In the “African Meningitis Belt,” the Harmattan seasons are associated with MeM outbreaks; similarly, the Santa Ana winds (SAWs) seasons are characterized by hot and dry winds (similar to Harmattan seasons) that occur seasonally in Southwest California, USA, and Northwest Baja California, Mexico. Objectives: We aimed to determine a potential association of SAWs with MeM in Tijuana, Baja California, Mexico, which in turn may partially explain the high rate of this disease in the region. Methods: Based on our previously published data obtained from thirteen years of active surveillance of MeM and a 65-year review showing the seasonal occurrence of SAWs, we estimated the risk ratio (RR) for the total case numbers of MeM (51 cases of children < 16 years old) vs. bacterial meningitis not caused by *Neisseria meningitidis* (NMeM, 30 cases, same age group) during seasons with and without SAWs. Results: We found an association between SAWs and MeM, but not with NMeM (RR = 2.06, *p* = 0.02 (95% CI 1.1 to 3.8), which may partially explain the high endemicity of this deadly disease in this part of the globe. Conclusion: This study shows a new potential climatic association with MeM and provides more information that justifies universal meningococcal vaccination in Tijuana, Mexico.

## 1. Introduction

Meningococcal disease (MD) is a contagious condition of *Neisseria meningitidis* transmitted from individual to individual by airborne respiratory droplets or throat secretions. MD transmission rates vary according to age, with children showing a lower rate of about 4.5% and persons older than 50 showing transmission rates of nearly 8%. Higher transmissibility rates are seen in adolescence and early maturity, with almost 10% of the population affected, though with several variations among regions [1]. Aggressive meningococcal infection is accountable for a broad clinical spectrum, and usually, signs and symptoms occur within 1–14 days following infection, although they often appear in the first 7 days. Early symptoms are similar to other bacterial infections, making prompt recognition problematic. Fever, nausea, vomiting, abrupt commencement of headaches, photophobia, neck stiffness, and altered mental state are typical symptoms [2]. Less common symptoms are pneumonia, conjunctivitis, otitis media, epiglottitis, urethritis, arthritis, and pericarditis. Meningococcal meningitis (MeM) occurs in 50–90% of cases. Nevertheless, in older people and immunodeficient hosts, pneumonic presentation appears more often, with serogroup Y as the foremost cause of pleuropulmonary complications [2].

Severe presentation is characterized by septicemia, or meningococcemia, which typically presents with a sudden onset of fever followed by a purpuric rash. Fulminant meningococcal sepsis can be developed in a few hours and might not have signs or symptoms of meningitis, accounting for 5–20% of cases. The disease can advance swiftly to septic shock, acute adrenal hemorrhage (Waterhouse–Friederichsen syndrome), and ultimately multi-organ failure [3].

Even with aggressive and proper treatment, the mortality rate of MD is between 10–25%. Moreover, the sequelae can be as harmful as the illness and may occur in 11–34% of survivors [3].

In Tijuana, Mexico, following 11 years of active surveillance, MD mortality among children and adolescents < 16 years of age was 25.5%, and, among survivors, 34% developed sequelae (mostly neurologic)^,^ which resulted in an economic burden average per case of USD 20,195, including both healthcare and societal costs, respectively [4].

The gold standard for diagnosing invasive meningococcal disease is the isolation of *N. meningitidis* from sterile body fluids (blood or cerebrospinal fluid (CSF)) or purpuric skin lesion scrapings. Given that meningococci can be a component of normal nasopharyngeal flora, their isolation from this site does not definitively confirm a clinical diagnosis of MD [1,2,3].

Furthermore, it is imperative to mention that when parenteral antibiotic treatment is initiated, the isolation rate of meningococci from blood culture drops from 50% to <5%, and the likelihood of CSF positivity via culture or microscopy is also rapidly reduced. Methods based on a rapid polymerase chain reaction (PCR) can complement standard laboratory procedures as they are less affected by prior antibiotic therapy, and these methods are being used increasingly [1,2,3].

Notwithstanding, MD is a vaccine-preventable condition [1,2,3]. The available meningococcal conjugate vaccines have an excellent safety profile. None have been associated with serious adverse effects, either during clinical trials or post-marketing surveillance.

For countries where endemicity has been established and reported, the WHO has expressed a solid recommendation to introduce appropriate largescale meningococcal vaccination programs based on the following criteria: countries with high (>10 cases/100,000 population/year) or intermediate endemic rates (2–10 cases/100,000 population/year) of MD and countries with frequent epidemics [5]. In these countries, the vaccine may be administered through routine immunization programs. However, supplementary immunization activities, for example, during outbreaks or through private vaccination services, are also recommended. Countries should select and implement the most appropriate control policy depending on the national epidemiology and socioeconomic resources [3,5].

In countries where the disease occurs less frequently (<2 cases per 100,000 population/year), meningococcal vaccination is recommended for defined risk groups, such as children and young adults residing in closed communities, e.g., boarding schools or military camps [3,5]. Laboratory workers at risk of exposure to meningococci should also be protected. Travelers to high endemic areas should be vaccinated against the prevalent serogroup(s) in these areas. Lastly, all individuals suffering from immunodeficiency, including asplenia, terminal complement deficiencies, or advanced HIV infection, should also be contemplated for vaccination [5].

When using conjugate vaccines, one recommended approach is to initiate a mass vaccination campaign for all children and adolescents aged nine months to eighteen years, followed by including the vaccine in the routine childhood immunization program [5]. Depending on surveillance data, other age groups could be included in the mass vaccination efforts: in the African meningitis belt, the broad age group of 1–29 years is the target for meningococcal A conjugate vaccination [5]. An alternative strategy would be to use conjugate vaccines for mass vaccination followed every 3–5 years for age groups at particular risk, as dictated by continued surveillance [5].

Knowledge and awareness of the MD burden are essential for all countries to tailor appropriate health policies for using available vaccines [2,3]. Countries considering implementing meningococcal vaccines should develop surveillance systems to characterize meningococcal disease epidemiology. They should also have a standard clinical case definition, field investigation of cases and outbreaks, and build laboratory capacities to confirm and characterize *N. meningitidis* [3,5]. Continued surveillance of MD should dictate the need and timing of repeat mass vaccination campaigns, as well as give an understanding of all the factors associated with this potentially lethal illness [5].

While *Neisseria meningitidis* serogroup A was the main cause for large epidemics before the monovalent conjugate serogroup A meningococcal vaccine was universally administered, serogroups W, C, and X are also responsible for localized outbreaks [6].

The highest burden of the disease occurs in the ‘African Meningitis Belt’, a region stretching from Senegal to Ethiopia with an estimated population of 400,000 million people [6]. In this African region, an increase in incidence is typically observed during the Harmattan seasons, characterized by a dry season between January to March, in which high dust loads are carried from the northeast before the rainy season commences [7]. These dust clouds mostly begin after the Bodélé depressions (the lowest geographical point in Chad). Following their appearance, it has been reported to cause weekly incidence rates of meningococcal disease to reach up to 100 per 100,000 of the population in individual communities [2]. Even with appropriate treatment, the mortality rate fluctuates around 10%, and 10–15% of survivors suffer long-term neurological sequelae [6,7].

Likewise, the Santa Ana winds seasons (SAWs) are episodic pulses of easterly, downslope, offshore flows over the coastal topography of the California Border Region, composed of Southern California (USA) and Northern Baja California (Mexico), occurring primarily from October to April, and are associated with dry air, often with anomalous warming at low elevations, similar to the Harmattan winds associated with MeM outbreaks in Africa [8]. A geographic representation of Harmattan and SAWs can be observed in Figure 1.

Equally important, SAWs have been associated with deleterious health conditions, such as episodes and/or exacerbations of asthma and/or allergies [9], and potentially with the spread of Coccidioidomycosis (a fungal disease transmitted by whirlwinds of air and soil) [10].

Our active national surveillance studies show that Tijuana has the highest incidence of MeM in Mexico. Yearly attack rates are 2.6 per 100,000 in all children and adolescents younger than 16. However, rates increase in younger populations, reaching 7.6 per 100,000 in infants and toddlers below two years old. Furthermore, *N. meningitidis* is the leading cause of all bacterial meningitis in the region, with meningococcal serogroup C being the predominant (61%), followed by Y (23%) [11,12,13].

Moreover, a binational San Diego, California, USA, and Tijuana, Mexico study between 2005–2008 showed that MeM was more frequent only at the General Hospital of Tijuana when compared to the whole of San Diego County in children < 16 years old (16 vs. 13 cases, respectively) [14].

On top of that, a serogroup C, MeM (clonal complex 11) outbreak occurred in Tijuana in 2013, with 19 cases, and it was associated with a high lethality rate of 36.8% [15].

In this study, we hypothesized that the high incidence of MeM in Tijuana might be partly associated with the occurrence of SAWs in the region. This association has never yet been described.

## 2. Methodology

To be able to formulate an association between SAWs and MeM, our first step was to display the MeM and NMeM cases admitted throughout time at the General Hospital of Tijuana. All meningitis cases identified in neonates were excluded from our analysis since early neonatal bacterial meningitis (first seven days of life) can be transmitted either vertically or horizontally from mother to infant.

According to one peer-reviewed publication from our group based on 13 years of active/prospective surveillance (2005–2018), in children >7 days and <16 years of age admitted with confirmed bacterial meningitis at the General Hospital of Tijuana, we identified 51 cases of MeM (25% lethality) and 30 NMeM, among which *S. pneumoniae* was predominant (42%), followed by *S. agalactiae*, *S. pyogenes*, *Enterobacteriaceae*, and others [13]. Additionally, MD cases without meningitis were also not included in our analysis; nonetheless, 92% of all MD patients developed meningitis [13].

The reason for comparing MeM with NMeM was basically to see whether only MeM, but not NMeM, was associated with SAWs, as with meningococcal outbreaks associated with the Harmattan condition in the African meningitis belt [7].

The second procedure to associate MeM with SAWs was to define a SAW episode. Consequently, based on a prior publication [8], it is defined as periods with at least 12 h of continuously selected winds that exceed the local wind speed threshold for at least 1 h.

Thirdly, from the available data between 1948–2012 (65-year summary review, considered to be a referral for SAWs seasonality), 2056 SAW episodes were detected, with an average of 32 occurrences per year; hence, a mean annual frequency of SAWs was estimated [8]. This paper also describes the time during the year when SAWs will most likely be present, usually between October and April [8].

Lastly, the total number of cases of MeM and NMeM per month (from our 13 years of active/prospective surveillance data [13]) were plotted and compared with simultaneous occurrence with/without SAWs seasons from the 65-year review [8] and followed by a risk ratio (RR) estimation.

A *z* test was also used to compare the proportions of MeM during the SAWs seasons vs. non-SAWs seasons.

## 3. Results

The RR of MeM and NMeM with/without SAWs is shown in Table 1. As shown, there is a clear, statistically significant association between MeM and SAWs compared to NMeM (RR = 2.06, *p* = 0.02, 95% CI 1.1 to 3.8).

Figure 2 shows individual cases per month of MeM (red dots) and NMeM (yellow dots) obtained from our 13 years of active surveillance, and, between the dotted lines, in blue bars, months with the highest mean annual frequency of SAWs, based on 65 years (1948–2012) of climatic surveillance and summary review [8]. This figure better illustrates how individual cases of MeM occurred more frequently during the periods of SAWs than NMeM, as statistically proved in Table 1.

Additionally, from 51 MeM cases, 44 (86.27%) occurred during SAWs seasons (*z* test = 7.32, *p* < 0.0002).

## 4. Discussion

This is the first study that clearly associates MD and a climatological condition other than the Harmattan seasons in the Sub-Saharan region. Both Harmattan and SAWs share similarities: (1) both are seasonal and cyclic; and (2) are associated with dry winds, and, in the case of Harmattan seasons, lead to dust loads that allow *N. meningitidis* to spread [7,8]. Nevertheless, the latter has never been described with SAWs. However, its impact on triggering allergies and asthma has been clearly described [9], and there is evidence that strongly suggests that these episodes of very dry air may be associated with a higher transmission of Coccidioidomycosis—a fungal disease that typically spreads contaminated soil during dry air turbulences [10].

In our study, we have shown, based on 13 years of prospective/active surveillance, that *N. meningitidis* is not the only leading cause of bacterial meningitis in Tijuana, Mexico [12,13]. Yet based on the same surveillance system in nine hospitals within the country, we found that MeM has much higher endemicity in Tijuana than in other geographic regions of Mexico [11], from which an explanation of this phenomenon is mandatory, aiding to further preventive strategies in Tijuana, such as vaccination.

Our elevated rates of MeM (2.69/100,000 in <16 years of age [12]) in a city of two million people may be seen as not concordant with fifty-one cases in thirteen years; nonetheless, as we described this when we published our studies [12,13], in the state of Baja California, Mexico, during the time of our active surveillance, there were mainly five different health care systems: “Instituto Mexicano del Seguro Social” (IMSS), “Seguro Popular” (“Popular Insurance”), “Instituto de Seguridad Social para Trabajadores del Estado” (ISSSTE), “Instituto de Seguridad Social para Trabajadores de Baja-California” (ISSSTECALI), and “PEMEX” [16]. Accordingly, the General Hospital of Tijuana was the only one in care of patients belonging to the “Seguro Popular” health care system; therefore, attack rates by age groups were estimated using the population of “Seguro Popular” as a denominator. The potential disadvantage is that maybe some patients with MeM belonging to the “Seguro Popular” system were hospitalized in the private sector, potentially leading to underestimation.

Even though MeM is a condition requiring a mandatory report in Mexico, no other hospital other than the General Hospital of Tijuana has a well-established active surveillance system that includes clinical and laboratory-trained personnel. A routine detection and follow-up of every MeM potential case are performed, and laboratory resources (cultures, PCR) are available to confirm or rule out each case. More details can be found in our previous publications [12,13].

Indeed, the main limitation of our study is the lack of data from other places where SAWs also occur, as this association is coming only from one hospital. However, in Baja California, Mexico, the General Hospital of Tijuana, where all our cases were obtained for this study, is the only hospital that performs active surveillance in the state (as mentioned). Hence, information from other hospitals where SAWs occur in the region is practically absent.

In Southern California, there are no studies of this association. Moreover, since universal meningococcal vaccination has been part of the USA National Immunization Program since 2005, trying now to make an association of MeM with SAWs could be very difficult to estimate. The latter can be inferred since, as we mentioned, we have published a binational (Tijuana, Mexico-San Diego, California) study between 2005–2008, having more MeM cases only at the General Hospital of Tijuana when compared to all the San Diego County [14].

A second limitation, as stated before, is the potential underestimation of either MeM or NMeM cases, since not all patients belonging to the “Seguro Popular” health system attend the General Hospital of Tijuana and may attend a private hospital.

We are aware that several other factors can be associated with the higher endemicity of MeM in our region. These factors include higher nasopharyngeal carriage in the population (studies never performed to date), and a high migration rate from Central and South America, the Caribbean, and even Africa or Eastern Europe, since migration and crowding are well-known risk factors for meningococcal outbreaks [17], highlighting that Tijuana, Mexico–San Diego, California, USA, is the highest transited frontier on the planet [18]. Nevertheless, our study shows a clear association between MeM and SAWs, a finding that helps us partially explain the higher endemicity of this potentially lethal disease in Tijuana compared to other regions in Mexico.

## 5. Conclusions

This report showed a statistically significant association of meningococcal meningitis during the Santa Ana winds seasons with bacterial but non-meningococcal meningitis, strongly suggesting that these hot, dry seasonal winds may stimulate a higher transmission and regular occurrence of this potentially deadly disease. Nonetheless, our results do not prove causality but instead support a robust hypothesis that, along with previous data showing a higher rate of MeM in Tijuana, Baja California, Mexico, urges serious consideration for universal vaccination against *N. meningitidis* in this region, and is aligned with WHO recommendations [5].

## Figures and Tables

**Figure 1 tropicalmed-08-00136-f001:**
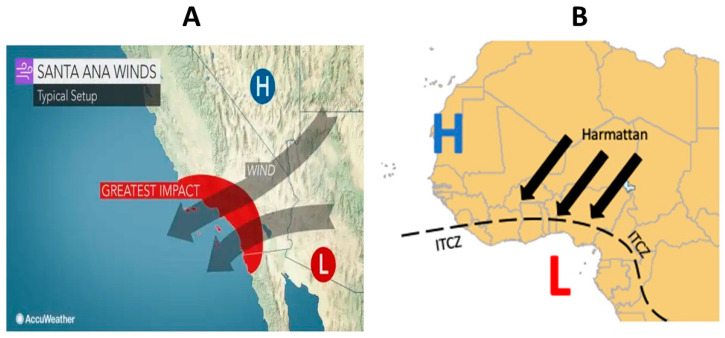
Geographical representation of Santa Ana winds in Southwest California, USA, and Northwest Baja California, Mexico (**A**); and African Harmattan seasons (**B**). Obtained from: https://www.accuweather.com/en/weather-news/what-are-santa-ana-winds-2/343027 (accessed on 17 February 2023); https://www.skybrary.aero/articles/harmattan (accessed on 17 February 2023).

**Figure 2 tropicalmed-08-00136-f002:**
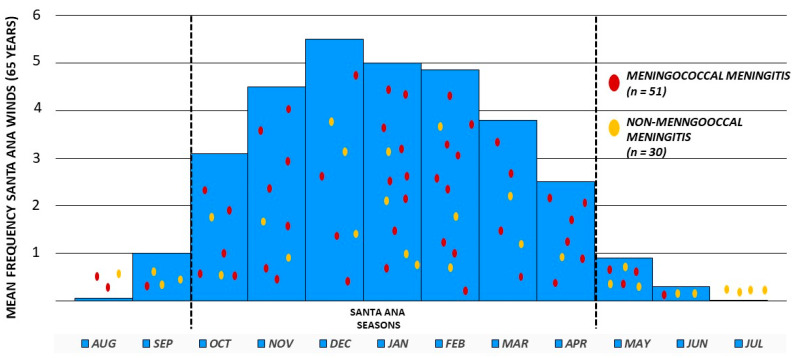
Association of mean annual frequency of Santa Ana winds (1948–2012) and meningococcal meningitis in Tijuana, Mexico, 2005–2018.

**Table 1 tropicalmed-08-00136-t001:** Risk ratio estimation of meningococcal meningitis (MeM) and bacterial non-meningococcal meningitis (NMeM) with Santa Ana winds (SAWs) seasons.

	SAWs Seasons	Non-SAWs Seasons
**MeM**	44 cases	7 cases
**NMeM**	17 cases	13 cases

**RR** = 2.06, *p* = 0.02 (95% CI 1.1 to 3.8).

## Data Availability

Not applicable.

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
