# Peer review of "Association between Meningococcal Meningitis and Santa Ana Winds in Children and Adolescents from Tijuana, Mexico: A Need for Vaccination"

_tropicalmed, 2023, doi:10.3390/tropicalmed8030136_

Round 1

Reviewer 1 Report

A table or a geomap with details of the seasonal Harmattan and Santa Ana Winds may be required for better clarity as audience from the remote east may not grasp the rationale.

Reviewer 2 Report

The brief report entitled "Association between Meningococcal Meningitis and Santa Ana 2 Winds in Children and Adolescents from Tijuana, Mexico, a 3 need for Vaccination", in which the authors studied a potential association of Santa Ana winds seasons with Meningococcal Meningitis in Tijuana, Baja-California, Mexico, which in turn explain the high rate of this disease in the region.  This study shows a new potential climatic association with Meningococcal Meningitis and provides more information that justifies universal meningococcal vaccination in Tijuana, Mexico. The study is interesting. There are a few minor issues that need to be fixed before considering the manuscript for publication. 

1. The abstract should be rewritten for better understanding and easy to follow. 

2. In the discussion section, discuss in detail about other potential factors associated with higher endemicity of Meningococcal Meningitis in Tijuana, Baja-California, Mexico with references. 

3. The given data regarding Meningococcal Meningitis are based on 13 years of active/prospective surveillance from 2005 to 2018. What is the current status (from 2018-2023) in these regions? 

Reviewer 3 Report

Please Find my comments the attached file. 

Round 2

Reviewer 3 Report

Dear authors thank you for your efforts to improve the quality of the manuscript.

I found that you unswered to almost all my comments. However, I still have some concerns:

1. In the introduction you added a long paragraphs (we tend to think that we are reviewing a new manuscript). Also, (paragraph 1 has no reference, while in the four remaining paragraphs you used only one reference (lines 71-106).  1 reference for 35 lines!!!! please try to summarize.

2. I still have a concern about the order of the figure and the table. since you did not add the descriptive data (that I have requested) and the redundancy of your ideas, I  suggest to return to the order of the first table (delete the term figure 2 from the methods)  

3. Line 14-15: delete the added sentence (To potentially...region) (you have included this in the objectives).

4. Line 15-18: try to relate the two sentences please.

5. Line 257: delete the point and the upper case for (Hilighting) to complete your sentence.

Author Response

We have also attached a file.

RESPONSES TO REVIEWER 3 SECOND ROUND

“Dear authors thank you for your efforts to improve the quality of the manuscript.”

Response: We deeply thank the reviewer for acknowledging our efforts to make our study of better quality.

“I found that you answered to almost all my comments. However, I still have some concerns:”

“1. In the introduction you added a long paragraphs (we tend to think that we are reviewing a new manuscript). Also, (paragraph 1 has no reference, while in the four remaining paragraphs you used only one reference (lines 71-106).  1 reference for 35 lines!!!! please try to summarize.”

Response: Indeed, the reviewer is right. Based on the fact we need to have at least 2,500 words required for this short report, we texted without adding the proper references, appropriately. Accordingly, we have added the required references.

“2. I still have a concern about the order of the figure and the table. since you did not add the descriptive data (that I have requested) and the redundancy of your ideas, I  suggest to return to the order of the first table (delete the term figure 2 from the methods) “

 Response: We apologize with the reviewer for not adding all descriptive data from all patients since, as we stated, is data already published (ref 13) and may deviate the readers´ attention to our main objective of this study. We have deleted the text regarding figure 2 from the methods section, and kept it after table 1 in the Results section, as kindly suggested by the reviewer.

  1. Line 14-15: delete the added sentence (To potentially...region) (you have included this in the objectives).

Response: I think the reviewer refers to the word “to partially explain…” in the abstract section, we have removed that word, thank you.

  1. Line 15-18: try to relate the two sentences please.

Response: Completely agree with the reviewer, we have made the required changes to relate both sentences (we added “similarly”).

  1. Line 257: delete the point and the upper case for (Hilighting) to complete your sentence.

Response: The reviewer is right again; corrections have been made accordingly (line 261).

Again, we appreciate all the reviewer comments and suggestions, as well as the time invested in making our study a much better one. Our gratitude always.

Dr. Prof. Enrique Chacon-Cruz
